# Subthreshold Effects of Low-Frequency Alternating Current on Nerve Conduction Delay

**DOI:** 10.3390/biomedicines13040954

**Published:** 2025-04-13

**Authors:** Michael Ryne Horn, Nathaniel Liam Lazorchak, Usama Kalim Khan, Ken Yoshida

**Affiliations:** 1Department of Biomedical Engineering, Indiana University—Purdue University Indianapolis, Indianapolis, IN 46202, USA; 2Weldon School of Biomedical Engineering, Purdue University, West Lafayette, IN 47907, USA; nlazorch@purdue.edu; 3Department of Neurology, Indiana University School of Medicine, Indianapolis, IN 46202, USA; usamkhan@iu.edu; 4Purdue Indianapolis Division, Purdue University, Indianapolis, IN 46202, USA

**Keywords:** low-frequency alternating current block (LFACb), nerve conduction modulation, action potential conduction delay, functional electrical stimulation (FES), sodium channel inactivation

## Abstract

**Background/Objectives:** Low-frequency alternating current (LFAC) has been shown to induce nerve conduction block (LFACb). However, the effects of LFAC on conduction delay prior to block remain unclear. This study investigates the impact of LFACb on conduction velocity and blocking thresholds in myelinated and unmyelinated fibers using experimental and computational models. **Methods:** Four models were employed to analyze LFACb effects: (1) in-vivo experiments in earthworms examined conduction delays across nerve bundles with distinct conduction velocities; (2) ex-vivo experiments in canine vagus nerves assessed the upstream and downstream effects of LFAC waveforms ranging from 50 mHz to 500 mHz; (3) in-silico simulations using the Horn, Yoshida, and Schild (HYS) model for unmyelinated fibers explored size-dependent conduction delays and blocking thresholds; and (4) in-silico simulations using the McIntyre, Richardson, and Grill (MRG) model extended to 504 Nodes of Ranvier characterized myelination effects, localized nodal interactions, and diameter-dependent thresholds. **Results:** LFAC-induced conduction delays were independent of LFAC frequency but strongly influenced by fiber diameter and conduction velocity. Larger fibers exhibited lower block thresholds and shorter delays before block onset. In contrast, smaller fibers demonstrated prolonged subthreshold conduction delays before achieving full block. **Conclusions:** These findings suggest that LFACb could serve as a neuromodulation tool for selectively blocking larger fibers while preserving smaller fiber function. This has potential applications in functional electrical stimulation (FES) and temporary, non-destructive nerve blocks for clinical and research applications.

## 1. Introduction

In recent years, research into low-frequency alternating current block (LFACb) and activation (LFACa) has been explored as a promising tool for neuromodulation. LFACb has been shown to effectively block autonomic nerve fibers at lower current thresholds compared to the kilohertz frequency alternating current (kHFAC) block, with the added advantage of avoiding the onset activation typically associated with the kHFAC [1,2]. Initial studies demonstrated that LFACb could reversibly block vagus nerve conduction in rat and porcine models, with the latter requiring higher currents (1.1±0.3 mAp) to achieve block due to its larger diameter [3].

While kilohertz-frequency alternating current (kHFAC) block remains a widely studied method for neuromodulation, requiring amplitudes as high as 12–35 mA at 20 kHz for similar-sized nerves, it is associated with onset activation, a limitation that has driven efforts to find alternative conduction block strategies [4,5]. Direct current (DC) block has been explored as another alternative, where a sustained DC field induces nerve conduction block through sodium channel inactivation via cathodic block [6]. Although DC block avoids the prolonged onset activation seen in kHFAC, it can still produce make-and-break excitations at the onset and offset of the waveform. Additionally, electrode interfaces tend to exhibit their highest impedance at DC, decreasing as frequency increases. This elevated impedance can result in large voltages even when delivering low current levels, which increases the risk of exceeding the water window and causing hydrolysis, leading to both tissue and electrode damage.

Efforts to reduce kHFAC-onset activation by superimposing DC offsets have shown some promise but remain limited in terms of their clinical applicability [7]. In contrast, low-frequency alternating current block (LFACb) achieves conduction block through the same fundamental mechanism as DC, sodium channel inactivation, but applies it through a sinusoidal, charge-balanced waveform. This alternating waveform avoids both the onset excitation seen in kHFAC and the make-and-break activation inherent to constant DC, resulting in a clean and reversible block with no unintended activation. The use of a slowly varying sinusoid that starts and ends at zero crossings further ensures smooth transitions in and out of block. While LFACb operates at low frequencies that approach DC-like behavior, its inherent charge-balancing effects, biphasic symmetry, and slightly reduced impedance make it functionally safer—particularly when paired with high-capacitance electrode materials such as platinum black or Poly(3,4-ethylenedioxythiophene) (PEDOT). These materials expand the charge injection window and mitigate the risk of hydrolysis at low frequencies.

This subthreshold conduction delay, while subtle, may be clinically relevant in scenarios where signal timing is crucial—such as coordinating motor and sensory pathways in rehabilitation, delaying afferent reflex loops, or desynchronizing pathological activity in tremor disorders. Unlike full conduction block, delaying conduction while preserving signal fidelity may enable more physiologic neuromodulation strategies, especially in functional electrical stimulation (FES), neuromotor rehabilitation, or temporary sensory modulation.

Conduction delay prior to full block, first observed during the initial discovery of LFACb [8], may offer valuable insight into the underlying mechanisms of this waveform. Understanding and harnessing this subthreshold behavior is critical for optimizing LFAC waveforms for clinical applications, such as functional electrical stimulation (FES) or pain management.

This study aims to address key gaps in the understanding of LFACb:Characterize the conduction delays induced by subthreshold LFAC in both unmyelinated and myelinated fibers.Investigate the size-dependent blocking dynamics of LFAC across fiber diameters.

By leveraging a combination of in-vivo, ex-vivo, and in-silico models, this study seeks to provide a comprehensive analysis of LFACb, shedding light on its mechanisms and potential applications in neuromodulation.

## 2. Materials and Methods

### 2.1. In-Vivo Earthworm Experiment

Earthworms was anesthetized via immersion in a 10% ethanol solution oxygenated with ambient air for 10 min at room temperature [9]. Earthworm length ranged between 100 and 120 mm. Once anesthetized, they were placed on a dissection tray, and a custom pentapolar electrode—constructed from platinum foil coated with platinum black—was positioned around the earthworm. Bipolar pulse stimulation was applied using two pins placed 20–40 mm towards the distal end and spaced about 5 mm apart. Three pins, spaced about 5 mm apart, were positioned 20–40 mm proximal to the pentapolar cuff electrode for a tripolar recording configuration, as shown in Figure 1. The tripolar recording setup was connected to a Gould bioelectric amplifier via a Windograf 980 (Gould Electronics Inc., Einchstetten am Kaiserstuhl, Germany) with cutoff frequencies ranging from 10 Hz to 5 kHz. Pulse stimulation was delivered using a pulse stimulator (SD9, Grass Medical Instruments, Quincy, MA, USA) at a pulse duration of 0.5 ms and a frequency of 4.4 Hz. A pulse recruitment curve was generated to ensure activation of both the medial giant fiber (MGF) and lateral giant fiber (LGF).

The pentapolar cuff, used as the blocking electrode, applied the LFACb waveform through contacts two and four. The electrode had a contact-center-to-contact-center pitch (CEPc) of 3 mm, a contact-center-to-cuff-edge distance (CEDc) of 3.25 mm, and a contact width (CEW) or 0.5 mm. The waveform was delivered using an electrochemical interface (SI1287, Solartron Metrology, Bognor Regis, UK) configured in galvanostat mode, with a sinusoidal input provided by an arbitrary function generator. The current monitor output from the electrochemical interface was displayed on an oscilloscope, and the direct current (DC) offset of the waveform was manually zeroed by adjusting the DC offset of the sinusoidal input on the arbitrary waveform generator. The current monitor output of the electrochemical interface, along with the signals from the tripolar recording pins, was recorded using a National Instruments data acquisition (DAQ) board (USB-6221-BNC, National Instruments, Austin, TX, USA) and processed with Mr. Kick II (Mr. Kick II, Larsen, SMI Aalborg University, DK, USA) at a sampling rate of 20 kHz. Recordings were obtained using LFAC frequencies of 50 mHz and 100 mHz.

### 2.2. Ex-Vivo Canine Vagus Experiment

Canine vagus nerves were extracted following postmortem trachea harvesting. The nerves were isolated from the connective tissue in the region from the middle cervical ganglia up to the mandible, length ranging 80 and 100 mm. The nerves were placed in an oxygenated Krebs–Henseleit (Composition in mM: NaCl, 113.0; KCl, 4.8; CaCl_2_, 2.5; KH_2_PO_4_, 1.2; MgSO_4_, 1.2; NaHCO_3_, 25.0; and glucose, 5.5) solution [10] to keep the nerve viable and prevent drying. All recordings were made at room temperature, 22±1 °C. The nerve was placed on a dissection tray and pinned on either end to stretch it to its original length.

Custom-built tripolar and bipolar cuff electrodes made of platinum foil coated with platinum black were placed on the vagus nerve. A bipolar stimulation cuff electrode was placed at one end of the nerve, followed by a series of three tripolar cuff electrodes spaced roughly equally through the remaining length of the nerve, roughly 10–20 mm between adjacent cuff electrodes. The first cuff was the upstream recording electrode, followed by the blocking electrode, and finally the downstream recording electrode. The bipolar stimulation cuff electrode was connected to an optoisolated stimulator (DS-3, Digitimer Ltd., Hertfordshire, UK) controlled by a pulse output on an arbitrary function generator set to a stimulation frequency of 12 Hz. The pulse width was set to 100 µs and the pulse amplitude was increased until the compound action potential (CAP) could be seen. A sinusoidal current waveform was delivered by the blocking cuff from a voltage-controlled current source (VCCS) (CS580, Stanford Research Systems, Sunnyvale, CA, USA) set by an arbitrary function generator. Contacts 1 and 3 were used to deliver the LFAC waveform, CEPc of 3 mm and CEDc of 1.75 mm. The VCCS had the shield set to return, floating isolation, and a gain of 1 mA/V. DC offsets were corrected manually with the adjustment knob prior to each recording. Both upstream and downstream tripolar recording cuff electrodes were connected to a variable gain invertible amplifiers (UIA 3.6, Yoshida, Indianapolis, IN, USA, 2020) [11]. The two recording cuffs, input sinusoid voltage, output sinusoid current monitor, and the stimulation cuff trigger were recorded at 20 kHz using the USB-6221-BNC DAQ and Mr. Kick II. Recordings were made with LFAC frequencies varying from 50 mHz up to 500 mHz.

### 2.3. In-Silico Canine Vagus Model

The ex-vivo canine experiments were replicated in-silico using computational simulations that incorporated both unmyelinated and myelinated nerve fibers. The simulations utilized Comsol Multiphysics (Comsol 6.2, Burlington, MA, USA) to model the volume conductor environment, while the nerve fiber models were implemented in Matlab (2023b, Mathworks Natick, MA, USA) for unmyelinated fibers and in NEURON (Version 8.0, Yale University) for myelinated fibers.

#### 2.3.1. Volume Conductor Model

A 2D axisymmetric finite element model (FEM) was constructed to simulate the electrical environment surrounding the vagus nerve. The nerve bundle was modeled as a cylinder with a 1.6 mm diameter containing a single fascicle of 1.3 mm diameter, consistent with prior characterizations of the vagus nerve’s structure [12]. The model included a 50 µm saline-filled gap between the epineurium and the inner diameter of the cuff electrode. Electrical properties for the endoneurium, perineurium, epineurium, and saline were based on previously reported conductivity and permittivity values (Table 1) [12,13].

The bipolar cuff electrode was configured with a contact edge width (CEW) of 0.5 mm, a contact-center-to-contact pitch (CEPc) of 3 mm, and a contact-edge-to-cuff-edge distance (CEDc) of 1.75 mm. Using this setup, the FEM produced spatial-temporal extracellular potential distributions along a longitudinal line 1 µs from the inner perineurium wall. These distributions, or weighting functions, were used to simulate the extracellular potential ($V_{e}$) applied to nerve fiber models.

#### 2.3.2. Unmyelinated Fiber Simulations

The unmyelinated fiber simulations were performed using the Horn, Yoshida, and Schild (HYS) model [14,15], which accurately captures ionic currents and action potential propagation dynamics in unmyelinated axons. Fiber diameters of 0.5 µs, 1 µs, and 2 µs, representing typical autonomic fibers in the vagus nerve, were simulated at a temperature of 20 °C. Extracellular potentials derived from the FEM were converted to activation functions by calculating the second spatial derivative ($\frac{\partial^{2}\;V_{e}}{\partial \;z^{2}\;}$) combined with the second spatial derivative of the membrane potential ($\frac{\partial^{2}\;V_{m}}{\partial \;z^{2}\;}$) in a modified cable equation (Equation (Equation 1)).

Single fiber action potentials (SFAPs) were initiated at the distal end of the model at a frequency of 12 Hz and analyzed at recording sites proximal and distal to the blocking region. Conduction delay and block thresholds were determined by measuring the timing and attenuation of AP propagation across these sites.(1)$$\frac{\partial \;V_{m}}{\partial \;t\;}=\left[\left(\frac{\partial^{2}\;V_{m}}{\partial \;z^{2}\;}+\frac{\partial^{2}\;V_{e}}{\partial \;z^{2}\;}\right)/R_{a}-\Sigma I\right]/C_{m}$$
where

$V_{m}$—is the membrane potential.

$V_{e}$—is the extracellular potential.

ΣI—is the sum of ionic currents.

$R_{a}$—is the axoplasmic resistance.

$C_{m}$—is the membrane capacitance.

*z*—is the spatial variable.

*t*—is the temporal variable.

#### 2.3.3. Myelinated Fiber Simulations

Myelinated fibers were modeled using the McIntyre, Richardson, and Grill (MRG) model [16], implemented in NEURON and extended to include 504 Nodes of Ranvier (NoR) to match the spatial scale of the FEM-derived weight functions. Fiber diameters of 5.7 µs, 8.7 µs, 11.5 µs, and 16.0 µs were simulated, representing large-diameter fibers in the vagus nerve. Weight functions generated from the FEM were aligned to a central NoR to minimize variability due to fiber positioning. These functions were sampled at each node location and scaled to the extracellular potential $V_{e}$ using reciprocity principles for a 1A peak input.

Action potentials were initiated at the distal end of the fiber at a rate of 40 Hz, and the transmembrane potential ($V_{m}$) was recorded at proximal and distal nodes relative to the blocking electrode. Analysis focused on conduction delays, block thresholds, and the localized effects of LFAC on nodal dynamics. Saltatory conduction, specific to myelinated fibers, was assessed to evaluate the impact of LFAC on the nodes nearest the blocking cuff.

## 3. Results

### 3.1. Time-Triggered Plots

The data presented in this paper demonstrate a time-triggered view in which continuously collected time data are segmented and viewed with respect to a trigger point. The *x*-axis of these plots shows the triggered time window, displayed in milliseconds, in which 0 ms is the time in which stimulation was evoked. The *y*-axis shows the continuous recording time, the longer time scale in which the smaller triggered windows where extracted from, displayed in units of seconds. These time triggered data sets create a matrix of data which allow for a 3D view of the recorded data, such as the surf plot from Matlab (Matlab 2023b, Mathworks, Natik, MA, USA). From this view, all stimulation points will be aligned on the left-hand side of the plot and delays in conducting action potential (AP) can be easily viewed.

### 3.2. Observation in Earthworms

When an LFAC sinusoidal waveform was applied to the earthworm, there was phasic block of the MGF and LGF at low current amplitudes, ±200 µA, as seen in Figure 1. In B1 and B2 of this figure, there are regions in which complete block of both the MGF and LGF was achieved for 100 mHz and 50 mHz frequencies, respectively. Other regions contain partial blocks where only the MGF or LGF are blocked while the other CAP passed through the blocking cuff. Prior to each block and release of block the CAPs, there were delays in conduction due to the effects of the LFAC waveform. There was no noticeable increase or decrease in the delay when changing the LFAC frequency but there was a difference when comparing the delay between the MGF and LGF. Figure 1(A1,A2) shows the current monitor output of the electrochemical interface. These occurrence of these waveforms at such low frequencies show some nonlinearities associated with the electrode hitting the water window; however, the blocks remained reversible and DC drift was within ±25 µA.

### 3.3. LFAC Observations in Canine Vagus Nerve

When scaling up to a mammalian model, a second recording point was added to see if the LFACb waveform could effect CAP upstream to the blocking electrode. Figure 2 shows the experimental setup as well as the CAP recordings upstream and downstream. The effect of full block on the downstream recording electrode was preserved while the upstream recording shows the CAPs were mostly undisturbed. Again, as was seen in the earthworm model, there was a delay in conduction prior to block. Figure 3 shows the triggered time traces in a 2-D plot; the averages show that the disturbance to the upstream electrode had little to no effect on the shape and amplitude of the CAP, whereas the downstream electrode showed marked differences between the blocking and no block phases. Figure 4 shows the time-triggered plots of the range of observed frequencies aligned with the stimulus artifact at 0 ms. Frequency appears to not have any effect on the amount of delay a CAP experiences before becoming fully blocked.

### 3.4. In-Silico Canine Analogy for Unmyelinated Fibers

In-silico results were obtained to mimic the ex-vivo canine experiments for unmyelinated fibers using the HYS model. Figure 5 shows the experimental setup at the top, with observation points upstream and downstream of the blocking cuff. The upstream recordings (Figure 5A,B) reveal a slight modulation of the single-fiber action potential (SFAP), similar to the results seen in the ex-vivo upstream recordings (Figure 2 and Figure 3). In Figure 3, the stimulus artifact remains unchanged while the CAP is modulated, showing that the LFAC has an effect before the CAP passes the blocking cuff electrode; this is likely due to currents escaping the structure of the cuff electrode. In contrast, the downstream recordings (Figure 5C,D) show a slowing-down of conduction and eventual block due to the LFAC waveform.

The effects of LFAC frequency on conduction delay are shown in Figure 6A–C. Across frequencies (50 mHz, 100 mHz, and 250 mHz), there were no observable differences in the magnitude of the conduction delay. However, the effect of the diameter of the axon on the conduction delay is shown in Figure 6D–F, where smaller axons (0.5 µs diameter) exhibited greater delays and higher blocking thresholds compared to larger axons (2.0 µs diameter). Figure 7 further confirms the relationship between fiber diameter and LFACb threshold, with larger fibers requiring lower current amplitudes to achieve block.

### 3.5. In-Silico Canine Analogy for Myelinated Fibers

Simulations were conducted to investigate the effects of LFAC on myelinated fibers using the MRG model. The results for fiber diameters of 8.7 µs, 11.5 µs, and 16.0 µs are presented in Figure 8. Figure 8A–C depicts conduction delays for a fixed fiber diameter (11.5 µs) at LFAC frequencies of 250 mHz, 100 mHz, and 50 mHz. Consistent with unmyelinated fibers, the conduction delay was independent of frequency. Figure 8D–F highlights the size-dependent nature of the conduction delay, where larger-diameter fibers exhibited shorter delays due to their faster conduction velocities (CV). The amplitude-dependent behavior of the LFAC-induced block is shown in Figure 8G–I for a 11.5 µs fiber at 100 mHz. At lower amplitudes (0.40 mA), delays were observed but block was not achieved (G). As amplitude increased (0.70 mA and 1.0 mA), full block occurred, with longer durations occurring at higher amplitudes. The saltatory conduction mechanism in myelinated fibers introduced distinct nodal blocking dynamics. LFAC-induced block was localized to a specific NoR, particularly those closest to the blocking cuff. These localized effects underscore the importance of electrode placement and field distribution in targeting myelinated fibers.

## 4. Discussion

The goal of this study was to determine the effect of the LFACb waveform on conduction delay before a block was achieved. The results provided key insights into how LFACb affects conduction dynamics across different models and fiber types.

### 4.1. Frequency Independence

The earthworm model data, derived from the initial discovery of LFACb, highlighted important characteristics of a slower conduction rate. First, the frequency of the LFAC waveform appeared to have no effect on the magnitude of the conduction delay. This invariance to LFAC frequency was consistently observed in larger mammalian models, such as the canine vagus nerve (Figure 4), where frequencies ranging from 50 mHz to 500 mHz produced similar delays. The in-silico models further confirmed this behavior across both unmyelinated and myelinated fibers (Figure 6A–C and Figure 8A–C). These findings suggest that an LFAC frequency within the tested range primarily influences the timing of block onset but not the extent of conduction-slowing.

### 4.2. Size-Dependent Conduction Delay

The conduction delay was observed to depend on axonal caliber and conduction velocity. In the in-vivo earthworm model, the slower-conducting LGF axons (CV: 5–10 m/s) exhibited longer conduction delays before complete block compared to the faster-conducting MGF axons (CV: 15–30 m/s) (Figure 1) [17]. This pattern was similarly observed in the in-silico unmyelinated HYS model, where smaller-diameter axons exhibited proportionally longer delays before LFACb-induced block (Figure 6D–F). For myelinated fibers, the MRG model confirmed this trend, demonstrating a size-dependent conduction delay before block in larger-diameter fibers (Figure 8D–F).

This size-dependent effect is consistent with the principle that larger-diameter axons, which have lower axoplasmic resistance ($R_{a}$), are more sensitive to extracellular fields, resulting in lower block thresholds. However, at LFACb amplitudes near the block threshold, smaller fibers remain in the “subthreshold delay” phase for longer, as their higher $R_{a}$ makes them less susceptible to immediate conduction block.

### 4.3. Amplitude Effects and Subthreshold Behavior

LFAC amplitude played a critical role in determining the time window for conduction delay and block. When the LFAC amplitude was just above the blocking threshold, blocking occurred at the peak in the sinusoidal waveform, where the rate of change in amplitude was near zero. This allowed for a longer conduction delay window at subthreshold levels before full block was achieved. In contrast, when the blocking threshold was much lower than the peak LFAC amplitude, as observed for the 2 µs fibers (Figure 6F and Figure 7), block onset was faster, with a reduced window of conduction delay.

At lower LFAC amplitudes near the block threshold, more of the waveform cycle remained in the subthreshold range, leading to a longer conduction delay window before block. Conversely, when the amplitude is well above the block threshold, a larger percentage of the waveform is supra-threshold, rapidly driving the fiber into a blocked state with minimal delay. The amount of time an AP is held in a subthreshold state and the fiber diameter do not greatly affect the amount of conduction delay, HYS: Figure 6D–F and MRG: Figure 8D–F. However, the fiber type (HYS vs. MRG) has a great affect on the amount of conduction delay observed. This can be seen between the HYS model, where the conduction delay is several milliseconds long (Figure 6), and the MRG model, where conduction delay is closer to a single millisecond (Figure 8). This helps explain the observed conduction delays in the earthworm model, where slower-conducting LGF fibers exhibited greater conduction delays relative to faster MGF fibers (Figure 1).

### 4.4. Upstream Modulation Effects

Ex-vivo recordings in canine vagus nerves included upstream electrodes to monitor potential LFAC effects upstream of the blocking cuff. Small perturbations were observed in the upstream CAP at higher LFAC amplitudes (Figure 2(A1,A2) and Figure 3(A1)). Similar effects were observed in the in-silico models, particularly at observation points closer to the blocking cuff. As the observation point moved further from the cuff, the effects diminished, indicating that the perturbations were due to the current spread. Such a spread is expected for any waveform type at sufficiently high amplitudes and highlights the importance of electrode placement for targeted blocking.

### 4.5. Large-Fiber-First Blocking Dynamics

Both the earthworm model (Figure 1) and in-silico results demonstrated a large-fiber-first blocking effect, where larger-diameter fibers were blocked before smaller ones. For example, the MGF was blocked before the LGF in the earthworm model (blue-outlined region) and reactivated earlier during recovery (green-outlined region). This effect is explained by the cable equation (Equation (Equation 1)), where the reduced axoplasmic resistance ($R_{a}$) in larger fibers amplifies the effects of the LFAC waveform’s activation function. Myelinated fibers exhibited a similar behavior, with larger-diameter fibers (e.g., 16.0 µs) showing lower blocking thresholds than smaller fibers (e.g., 8.7 µs) (Figure 8G–I).

### 4.6. Mechanisms of Conduction-Slowing

Conduction velocity-slowing appears to be a result of sodium channel inactivation, a known characteristic of cathodic DC block [18]. This was evident in both earthworm (Figure 1) and canine (Figure 2) experiments, where the lowest thresholds were observed at the cathode. The in-silico results (Figure 7) further support this, showing that the block is primarily determined by the distribution of the extracellular field.

Figure 7 indicates minimal differences in block thresholds for cathode-first versus anode-first configurations for an unmyelinated HYS model, Figure 8G demonstrates an uneven effects on conduction delay for cathode-first versus anode-first configurations. This unevenness was also observed in-vivo (Figure 2, Figure 3 and Figure 4) and likely is a result of the alignment of the NoR’s. These findings suggest that cathodal effects dominate the LFACb mechanism, which is the same mechanism as a DC block.

As for the length of conduction delay, both in-vivo and in-silico experiments show that the slower unmyelinated fibers are able to sustain more of a conduction delay before a block is achieved. Figure 1 provides a great example, showing a difference in the length of conduction delay, where the slower LGF (CV: 5–10 m/s) is able to achieve around 3 ms of delay while MGF (CV: 15–30 m/s) only achieves delays of about 1 ms. This can be seen in-silico when comparing the unmyelinated HYS and myelinated MRG models in Figure 6 and Figure 8, respectively. In the HYS example, the conduction delay was found to be as large as 10 ms and the MRG models were much shorter, at closer to 1 ms.

### 4.7. Mechanisms of LFACb

Mechanistically, LFACb functions as a DC block by inducing sodium channel inactivation in a phase-dependent manner. However, unlike conventional DC block, which applies a sustained potential gradient, LFACb alternates polarity over time, creating a more dynamic extracellular field. This oscillatory nature helps mitigate, though not entirely prevent, issues such as electrode polarization and electrolysis, particularly at lower amplitudes. By maintaining a moving current with periodic charge return, LFACb represents a step toward a more sustainable neuromodulation approach while retaining the conduction block characteristics of DC stimulation.

### 4.8. LFACb Safety Concerns

When working with DC or near-DC frequency waveforms like LFACb, it is critical to consider the charge injection capacity of the electrode material and ensure that stimulation remains within the electrochemical water window. Exceeding the water window can lead to electrolysis, producing harmful byproducts such as hydrogen and oxygen gas, which may cause tissue damage and electrode degradation. Electrode surface modifications, such as platinum black or conductive polymers like poly(3,4-ethylenedioxythiophene) (PEDOT), have been shown to enhance charge injection capacity and reduce impedance, thereby improving the stability of electrochemical interfaces [19]. These treatments can help mitigate the effects of polarization and enable safer, long-term applications of LFACb and other neuromodulation techniques involving low-frequency electrical stimulation.

To ensure that stimulation remained within electrochemical safety limits and did not exceed the water window, both the applied current waveform and the resulting voltage were monitored in real-time. Voltage deviations or waveform distortions were used as indicators of the water window being exceeded, helping to assess the safe operating ranges for LFACb. While LFACb amplitudes were generally kept below this threshold, at lower frequencies (50 mHz and 100 mHz), achieving conduction block necessitated larger amplitudes that exceeded the water window, as evident in Figure 1. This highlights the inherent challenge of using low-frequency waveforms for nerve block, where a higher charge injection capacity is required for efficacy. This finding further underscores the importance of electrode coatings, such as platinum black or PEDOT, to improve charge injection capacity and mitigate potential electrochemical instabilities.

## 5. Conclusions

This study observed an effect of LFACb in which the CAP slows down prior to being fully blocked. It was determined that the frequency of the LFAC waveform has no major effect on the degree of conduction delay. Instead, the diameter of the fiber and the conduction velocity played more critical roles, changing not only the threshold of the block but also the amount of conduction delay that is possible before a block is achieved.

This study also demonstrated that LFACb preferentially blocks fast-conducting, larger-diameter fibers before slower-conducting, smaller-diameter fibers. This property could be applied to functional electrical stimulation (FES) to address a longstanding challenge in neurostimulation. Traditional pulse stimulation recruits large-diameter fibers first, leading to increased muscle fatigue compared to natural, size-wise recruitment patterns [20]. By combining LFACb with typical pulse stimulation, it may be possible to activate a nerve bundle fully and selectively derecruit large-diameter fibers with LFACb, mimicking physiological recruitment and reducing muscle fatigue during FES [21].

Additionally, LFACb offers potential as a non-destructive alternative to cryo nerve block. The effects of LFACb closely mimic those of local cooling, which blocks faster-conducting fibers first [22] and induces conduction-slowing prior to block onset [23]. Unlike cryo nerve block, which permanently damages nerves for localized pain reduction [24], LFACb provides a reversible, non-destructive method for nerve modulation. This positions LFACb as a promising tool for applications such as post-operative pain management and other scenarios requiring temporary nerve block.

This study also helped clarify the mechanism of LFACb as a cathodic block that inactivates sodium channels. With a better understanding of the underlying mechanism, the potential applications of LFACb in areas such as FES, nerve modulation, and pain management become clearer. While most prior studies focused on autonomic, primarily unmyelinated fibers [1,3], this study provides insight into mixed nerves, demonstrating that LFACb can be effective in blocking somatic fibers at lower currents. Future work should aim to further characterize these mechanisms and optimize LFAC waveforms for specific clinical applications.

Additional in-vivo studies are needed to strengthen these findings. The earthworm model, with its two distinct CAPs (MGF and LGF) exhibiting different conduction velocities, presents a unique opportunity for further exploration. Future studies could focus on determining the blocking thresholds for the MGF and LGF individually and testing whether a slightly suprathreshold LFAC waveform produces equivalent conduction delays for both fibers at the same frequency. If similar delays are observed, this would confirm that the shape of the waveform is the dominant factor. Conversely, if delays remain different, fiber diameter may play a more significant role. It is hypothesized that waveform shape is the primary factor and that frequency may influence the delay when the waveform is just at the suprathreshold, contrary to the results observed in this study.

Moreover, further simulations using a myelinated fiber model, such as the McIntyre–Richardson–Grill (MRG) model [16], could investigate whether slower-conducting axons exhibit larger conduction delays than faster-conducting axons under similar conditions. This would help validate the findings from unmyelinated fibers and ensure the broader applicability of the conclusions across mixed nerve compositions.

## Figures and Tables

**Figure 1 biomedicines-13-00954-f001:**
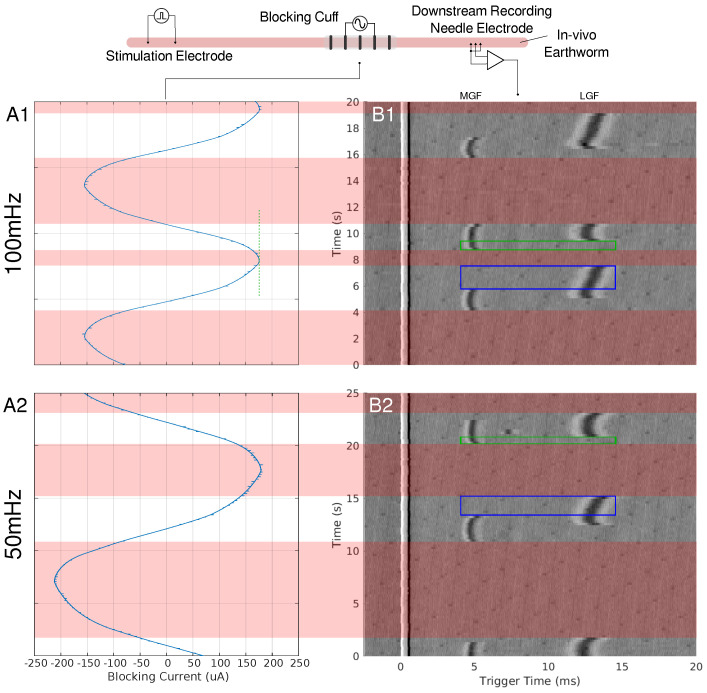
A depiction of the earthworm experimental setup can be seen at the top of the figure. Needle electrodes were used to stimulate the earthworms’ MGF and LGF nerve bundles. A pentapole cuff or blocking cuff, electrode was used to deliver the LFACb waveform. Needle electrodes were used in a tripolar configuration to make a recording of the effects of LFAC distal to the blocking cuff electrode. In column A, the current monitor out of the electrochemical interface or the delivered current is shown for 100 mHz (**A1**) and 50 mHz (**A2**) LFAC. Column (**B**), shows the resultant tripolar recorded time-triggered data for a pulse stimulation rate of 4.4 Hz. The stimulus artifact can be seen at the triggered time of 0 ms. The MGF and LGF CAPs can be seen around 5 ms and 12 ms, respectively. The areas shaded with red show the phases in which the LFACb waveform blocks both CAPs while smaller areas, outlined in blue and green, show areas of MGF block or LGF block, respectively.

**Figure 2 biomedicines-13-00954-f002:**
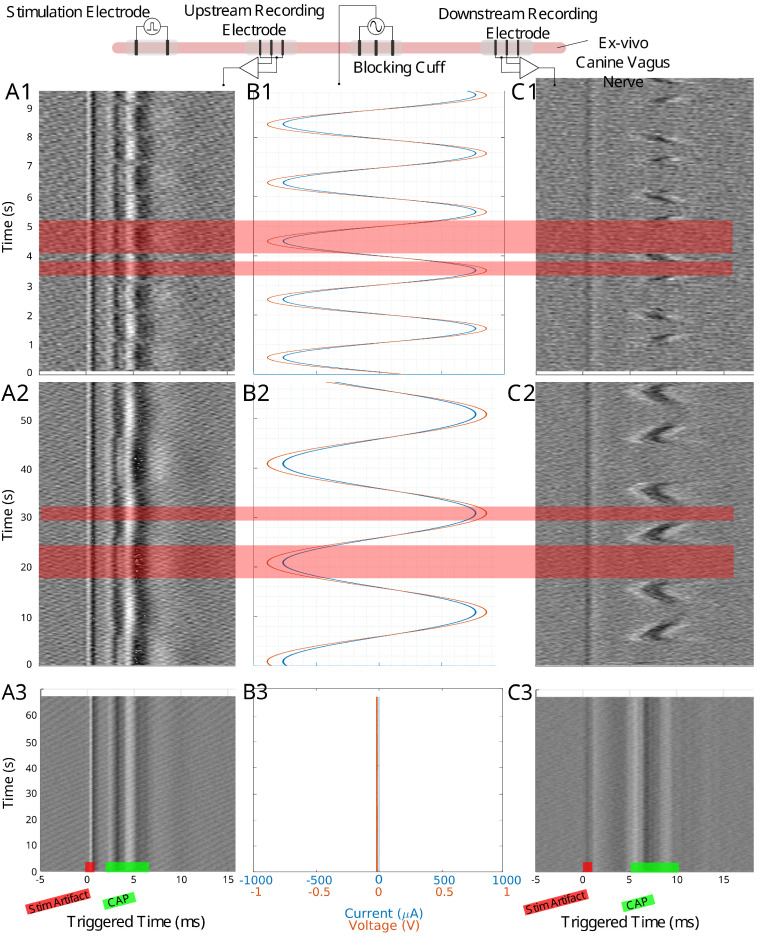
The experimental setup for the canine ex-vivo experiment is depicted at the top of this figure. An upstream bipolar cuff delivers a pulse stimulation to evoke a traveling CAP. The blocking cuff delivers the LFAC waveform to modulate the nerve activity while upstream and downstream tripolar cuff electrodes record the CAP activity. Column (**A1**–**A3**) shows the time-triggered data for the upstream tripolar recording electrode while column (**C1**–**C3**) shows the downstream tripolar recording electrode. Column (**B1**–**B3**) shows the current and voltage delivered by the blocking cuff. Row 1 shows data for a 500 mHz LFAC waveform, row 2 shows a 50 mHz LFAC waveform, and row 3 shows an example where there was no stimulation from the blocking cuff. The locations of the stimulus (stim) artifact and CAP can be seen in row 2.

**Figure 3 biomedicines-13-00954-f003:**
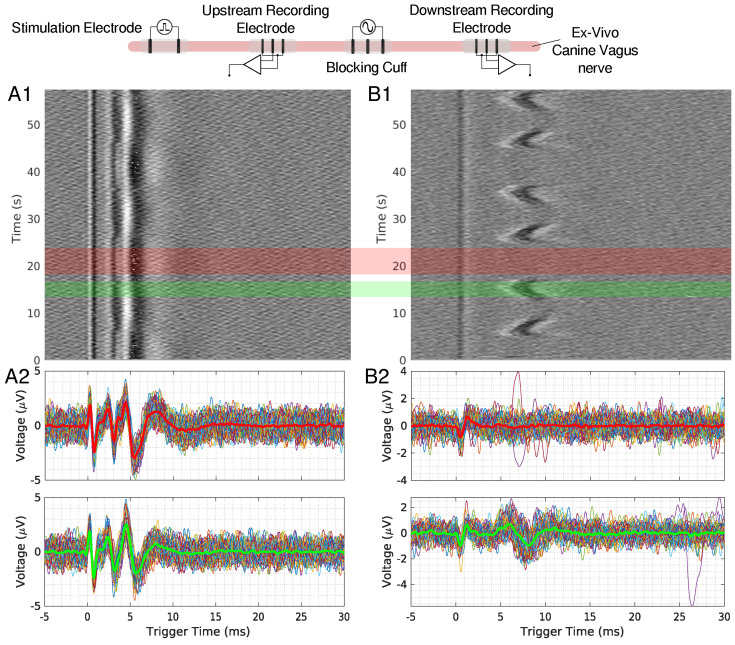
(**A1**,**B1**) show the time triggered data for an upstream and downstream 50 mHz LFAC ex-vivo canine experiment, respectively. Individual traces for regions with block (red region) and no block (green region) are shown in row (**A2**,**B2**) for upstream and downstream recordings, respectively. The thicker line in these plots represents the average of the extracted data, and red waveform represents the average of a block region, and green represents a no-block region.

**Figure 4 biomedicines-13-00954-f004:**
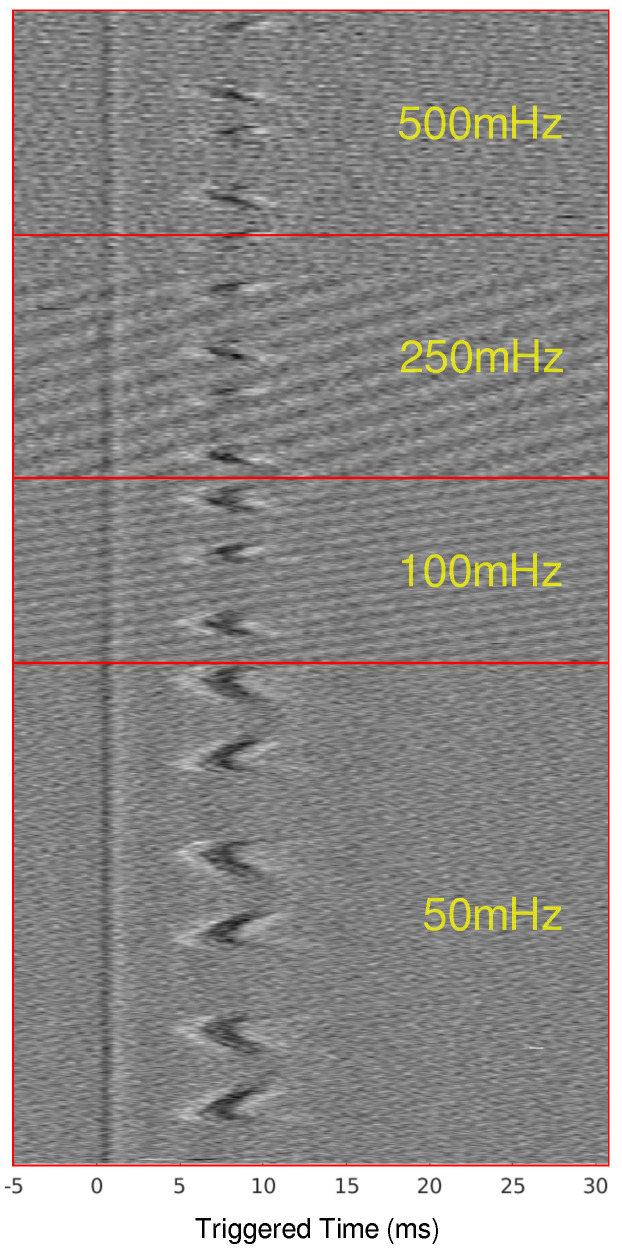
Time-triggered data for ex-vivo canine experiments for frequencies from 50 mHz to 500 mHz. The time-triggered data were aligned by the stimulus artifact to show the effect that frequency has on the total conduction delay before the CAP is blocked.

**Figure 5 biomedicines-13-00954-f005:**
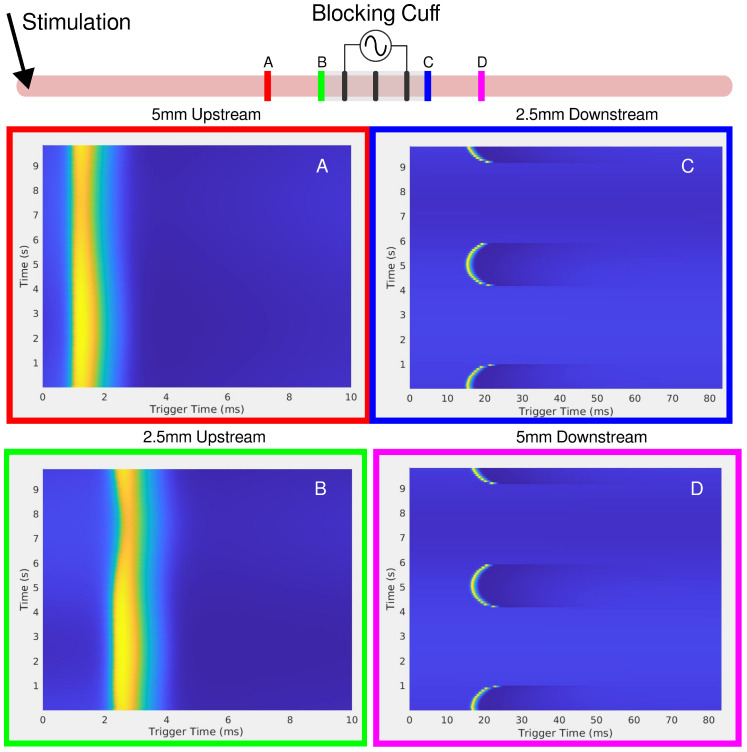
The in-silico setup is shown on the top of the figure as a cartoon. Stimulation of the axon occurs upstream, on the left-hand side. Four viewing points were chosen at distance of 5 and 2.5 mm from the center of the cuff, both upstream and downstream. (**A**) upstream 5 mm, (**B**) upstream 2.5 mm, (**C**) downstream 2.5 mm, and (**D**) downstream 5 mm. The red and green observation points on the left show the disturbance to the SFAP, which decreases further away from the blocking cuff.

**Figure 6 biomedicines-13-00954-f006:**
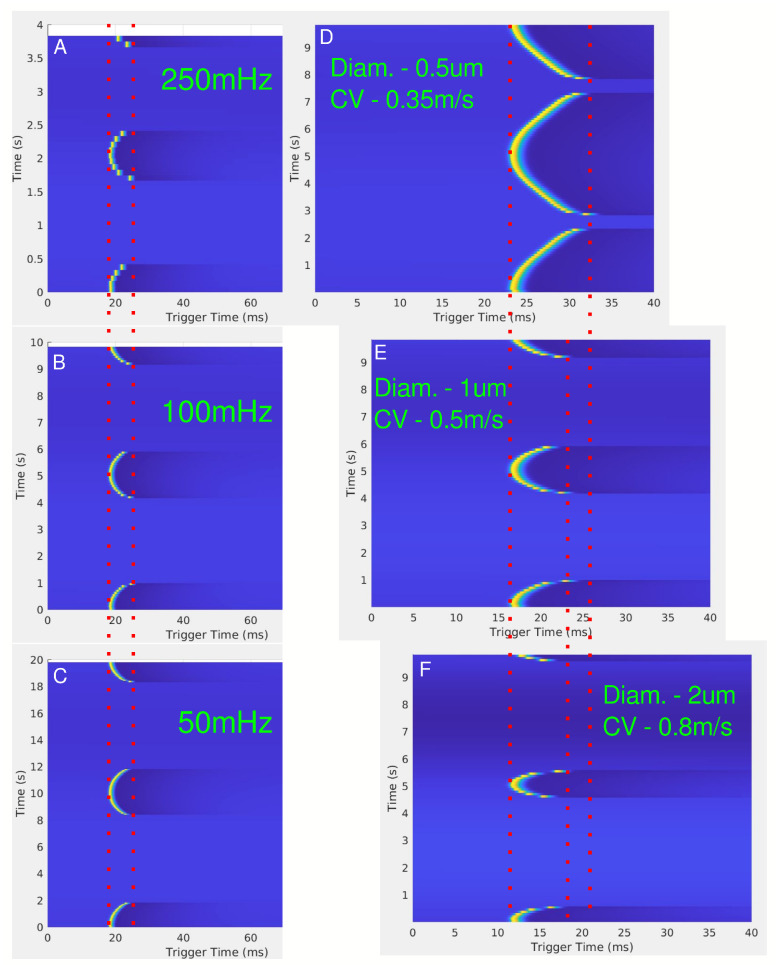
(**A**–**C**) show the effect of conduction delay for LFAC frequencies of 250 mHz, 100 mHz, and 50 mHz, respectively, for a 1 µs diameter unmyelinated HYS axon 5 mm downstream from the center of the cuff electrode. The red dotted lines show the range of the conduction delay. (**D**–**F**) show the effect of changing the axon diameter, or conduction velocity, for a frequency of 100 mHz. The red dotted lines are for visual aid for the SFAP delays.

**Figure 7 biomedicines-13-00954-f007:**
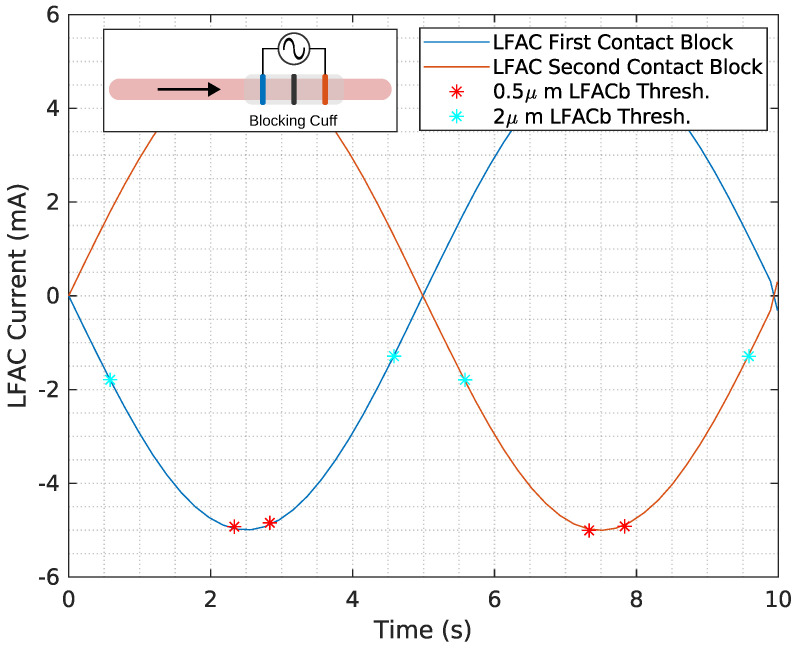
The LFAC current threshold is shown for 0.5 µs (red *) and 2 µs (cyan *) diameter unmyelinated HYS model for a frequency of 100 mHz. Cathodic LFACb occurs on both the first and second contact as the SFAP passes through, indicated by the direction of the black arrow. A depiction of the in-silico model can be found in the inset in the figure.

**Figure 8 biomedicines-13-00954-f008:**
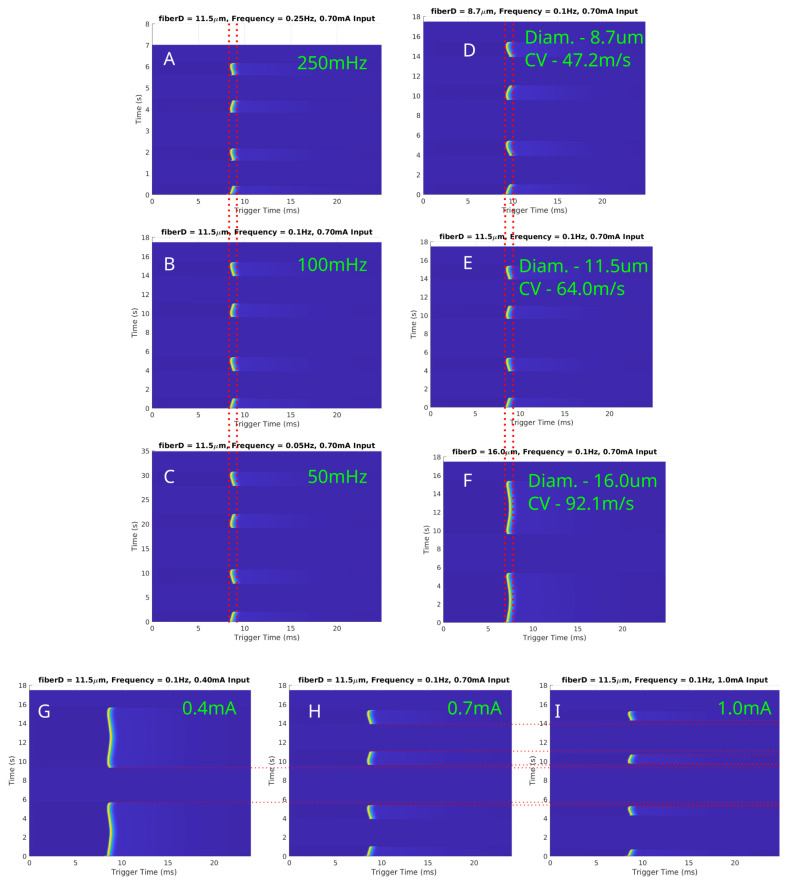
In-silico LFAC effects on myelinated fibers modeled using the MRG model at 150 nodes downstream of the center of the cuff electrode. (**A**–**C**) Conduction delays for a fixed fiber diameter (11.5 µs) at LFAC frequencies of 250 mHz, 100 mHz, and 50 mHz, respectively, demonstrate frequency independence. (**D**–**F**) Size-dependent delays for diameters of 8.7 µs, 11.5 µs, and 16.0 µs show shorter delays with increasing conduction velocities. (**G**–**I**) LFAC amplitude thresholds at 100 mHz for a 11.5 µs fiber indicate that increasing amplitude reduces block duration. Red dotted lines are for visual aid for comparing SFAP delays or block.

**Table 1 biomedicines-13-00954-t001:** Electrical Properties of the Nerve Bundle and Conductive Media.

	Endoneurium	Perineurium	Epineurium	Saline
Conductivity (S/m)	Radial: 0.083	$1\;\times \;10^{-3}$	$18\;\times \;10^{-3}$	2
	Longitudinal: 0.57			
Relative Permittivity (−)	75	2018	$9.4\;\times \;10^{6}$	75

## Data Availability

Data available upon request to the corresponding author.

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
