# Peer review of "Subthreshold Effects of Low-Frequency Alternating Current on Nerve Conduction Delay"

_biomedicines, 2025, doi:10.3390/biomedicines13040954_

Round 1

Reviewer 1 Report

Comments and Suggestions for Authors

This paper provides some new data about the properties of low frequency waveforms for nerve block.  However, the mechanism for block in these waveforms is fundamentally a DC block.  When the waveform passes above a direct current (DC) block threshold in either the cathodic or anodic phase, block is observed.  In the parts of the waveform near zero, the current is below block threshold and some level of partial block is observed in proportion to the current level.  Since DC is generally considered unsafe without appropriate mitigation methods, I can understand the reluctance of the authors to identify their technique as “DC”.  However, it is clear that the authors are aware of this issue.  The authors specifically use platinum black electrodes in order to extend the water window of the electrode.  However, they provide no details about the capacitance of these electrodes.  They also are monitoring the current output and observe, “These waveforms at such low frequencies show some nonlinearities associated with the electrode hitting the water window”.  So, they are clearly aware of the safety concerns associated with this technique.  These concerns need to be highlighted so that the reader understands the limitations of this technique.

Since the authors are avoiding mentioning DC as a mechanism, they neglect to cite a fundamental paper related to this work.  Bhadra, Kilgore 2004, “Direct Current Electrical Conduction Block of Peripheral Nerve”.  This paper provides mechanistic Neuron simulations and supports these simulations with in vivo experiments.  In particular, in the Horn paper (under review), the section “4.6 mechanism of conduction slowing” would benefit from a review of the Bhadra paper in regard to block thresholds and the effect of the DC field on the block properties.  The contribution of “virtual cathodes” described in the Bhadra paper is probably a factor in some of the conduction delays.  The voltage profile from the volume conductor model (in Horn paper) would give you a comparison graph to the outcomes in the Bhadra paper.  The Bhadra paper also describes DC blocking of large fibers at lower values which is a fundamental outcome of the paper currently under review.

Two issues that must be resolved for publication

  • It needs to be made clear to the reader that DC block is the mechanism producing nerve block.
  • Since DC has safety concerns, the authors have to describe and provide details about how they are mitigating this.

Some other clarifications:

  • Figure 1, 2 needs an example of the CAP with no block
  • The time triggered plots are good examples, but you need to quantify the amount of block (attenuation of signal) and the duration of the time delays or conduction velocity. These values can then be plotted vs frequency (figure 4), or frequency/diameter (figure 6 and 8). Figures 4, 6 and 8 are trying to show the relative delays and block thresholds, but it’s hard to see that by just looking at the examples side by side.  If you use conduction velocity instead of delay for these figures, be sure to make the text consistent with your metric.
  • In the discussion, the authors indicate that the delay is independent of the frequency. This is consistent with the mechanism being fundamentally DC block
  • In the discussion, the “size dependent conduction delay” is a little hard to follow, but if you make figures 4, 6 and 8 into plots of the delay you’ll be able to refer to those.
  • The amplitude and subthreshold behavior is also explained by DC as a mechanism. The steeper slope isn’t really the issue, it’s just that a larger percentage of the waveform is above the block threshold. 

Author Response

Comment 1: It needs to be made clear to the reader that DC block is the mechanism producing nerve block.

Response 1: We have cited the paper “Direct Current Electrical Conduction Block of Peripheral Nerve” in the introduction to provide context on DC block mechanisms. Additionally, we have clarified in the discussion (subsection Mechanisms of LFACb) that LFACb operates through the same fundamental mechanism as DC block.

Comment 2: Since DC has safety concerns, the authors have to describe and provide details about how they are mitigating this.

Response 2: We appreciate the reviewer’s concern regarding the safety of DC block. In response, we have added a subsection titled “LFACb Safety Concerns” in the discussion, which highlights the importance of electrode interfaces with high charge injection capacity. This section also details how we monitored the current and voltage inputs of the LFACb waveform in real time to detect nonlinearities, which could indicate exceeding the electrochemical water window.

Comment 3: Figure 1, 2 needs an example of the CAP with no block

Response 3: We appreciate the reviewer's suggestion to include an example of a CAP with no block for Figure 1. While we were able to add a no-block example to Figure 2 for the canine experiments, unfortunately, a no-block condition was not explicitly recorded for the earthworm experiment. The earthworm model was primarily used to demonstrate qualitative observations of conduction delay and block dynamics, and in this particular dataset, recordings were focused on LFAC conditions. However, the conduction velocities and waveforms for the medial giant fiber (MGF) and lateral giant fiber (LGF) have been previously well-characterized in literature (e.g., Shannon et al., 2014), which provides a reference for how these CAPs would appear in the absence of block.

We acknowledge that having a direct no-block condition for Figure 1 would strengthen the comparison, and we will consider incorporating additional controls in future experimental designs to facilitate direct quantitative analysis. However, for the current study, we believe the existing qualitative analysis is sufficient to illustrate the observed conduction delay and block characteristics.

Comment 4: The time triggered plots are good examples, but you need to quantify the amount of block (attenuation of signal) and the duration of the time delays or conduction velocity. These values can then be plotted vs frequency (figure 4), or frequency/diameter (figure 6 and 8). Figures 4, 6 and 8 are trying to show the relative delays and block thresholds, but it’s hard to see that by just looking at the examples side by side.  If you use conduction velocity instead of delay for these figures, be sure to make the text consistent with your metric.

Response 4: We appreciate the reviewer’s suggestion to quantify conduction block (attenuation of signal) and conduction velocity across different frequencies and fiber diameters. While this would indeed strengthen the analysis, the temporal resolution of certain datasets—particularly at higher frequencies, such as in Figure 4 (canine data)—limited our ability to perform precise quantitative measurements. Given these constraints, this study focuses on qualitative observations, highlighting the distinct phase-dependent effects of LFACb.

To address this, we incorporated in-silico models that mimic the in-vivo/ex-vivo conditions, demonstrating strong qualitative similarities between experimental and computational results. These findings suggest that a more extensive quantitative analysis could be conducted in a dedicated in-silico study, which could then be validated through targeted animal studies for calibration and verification. We acknowledge that this would be an important next step to fully characterize the relationships between frequency, conduction delay, and block thresholds in a controlled computational framework.

Comment 5: In the discussion, the authors indicate that the delay is independent of the frequency. This is consistent with the mechanism being fundamentally DC block

Response 5: We appreciate the reviewer’s comment. In response, we have dedicated a subsection in the discussion titled “Mechanisms of LFACb” to explicitly state that LFACb is fundamentally a DC block, reinforcing its frequency-independent effects on conduction delay.

Comment 6: In the discussion, the “size dependent conduction delay” is a little hard to follow, but if you make figures 4, 6 and 8 into plots of the delay you’ll be able to refer to those.

Response 6: To improve readability, we have revised this section in the Discussion to better delineate the observed trends between in-vivo, in-silico unmyelinated, and in-silico myelinated results.

Regarding the suggestion to create quantitative plots of conduction delay for Figures 4, 6, and 8, we agree that this would provide additional clarity. However, the data resolution in certain experimental conditions, particularly at higher frequencies (e.g., Figure 4, canine data), was not sufficient to extract precise numerical conduction delay values. Instead, we have emphasized the qualitative trends observed in both the experimental and computational models, which consistently demonstrate size-dependent conduction delay before block.

The strong alignment between in-vivo/ex-vivo and in-silico results suggests that a more controlled computational study could provide a comprehensive quantitative analysis of these effects. As a next step, a dedicated in-silico study could allow for detailed quantitative assessment, which could then be further validated in targeted animal models.

Comment 7: The amplitude and subthreshold behavior is also explained by DC as a mechanism. The steeper slope isn’t really the issue, it’s just that a larger percentage of the waveform is above the block threshold.

Response 7: We appreciate the reviewer's insightful comment regarding amplitude-dependent subthreshold behavior and its relation to DC block mechanisms. In response, we have revised the discussion of amplitude effects to emphasize that the primary factor influencing conduction delay is the proportion of the LFAC waveform that remains above the block threshold. Rather than attributing subthreshold behavior to the rate of change (slope) of the waveform alone, the revised text clarifies that when the block threshold is close to the LFAC amplitude, a larger fraction of the waveform cycle resides within the subthreshold range, leading to prolonged conduction delays before full block onset. Conversely, when the block threshold is well below the LFAC amplitude, block occurs rapidly, reducing the conduction delay duration.

Reviewer 2 Report

Comments and Suggestions for Authors

The study describes the effects of low-frequency alternating current (LFAC) on action potential conduction, focusing on conduction delays prior to LFAC block (LFACb) onset. The manuscript is generally well structured and clear. The methods and key characteristics are well described, with important findings highlighting LFACb as a promising tool for neuromodulation, offering potential applications in functional electrical stimulation (FES) and temporary, non-destructive nerve blocks. The present work requires some improvements before acceptance for publication.

Comments
1) Line 41: the sentence mentions three key gaps, however later on there are two enumerations.

2) 2.1. In-vivo Earthworm Experiment: indicate the length of the worm as performed with the canine vagus nerve and distances of the stimulation, blocking and recording sites.

3) 2.1. In-vivo Earthworm Experiment: why did you use a pentapolar cuff for blocking if you only required a cathode and anode?

4) For sections 2.1 and 2.2, specify the distances in millimeters for stimulation, blocking, and the upstream and downstream recordings. Providing these details will be valuable for other research groups working with similar in-vivo and in-silico models.

5) Lines 206-207: It is mentioned that, in the upstream recording cuff, the effect observed before the CAP passes the blocking cuff electrode is likely due to currents escaping the structure of the cuff electrode. To support this statement, have you measured the contribution of the current escaping from the cuff and affecting the membrane potential in the surroundings in your in-silico model? If this is the case, please add corresponding information in the Results section.

6) Lines 224–226: To improve visualization, consider adding a plot similar to Figures 1 and 2, where the maps are shown alongside the blocking waveform. A zoomed-in view of the waveform, focusing on this small current range, could be beneficial for the reader. Additionally, the phrase ‘As amplitude increased (0.70 mA and 1.0 mA), full block occurred with shorter durations at higher amplitudes,’ ensure clarity in describing the relationship between amplitude and blocking duration, because from Figure 8 G-I, as the current increases in amplitude the full block duration intervals increase.

7) In section 4.3, line 255, is it conduction delay? The reason is that in Figure 8 G-I, as the amplitude of the LFAC increases, the conduction delay does not show a change. Contrary is the time duration of blocking which increases with LFAC amplitude.

8) In line 289, Figure 6G is not present.

9) In section 4.6, lines 291-292, it says that ‘smaller diameter fibers are more sensitive to the local extracellular field…’. This statement is not fully clear considering that small fibers have a larger threshold which can be understood as less sensitive to extracellular fields. Please add more explanation. In addition, the unevenness is also not fully clear in images. See previous comment.

10) In the manuscript, I couldn’t find any mention on the different thresholds (cathodic and anodic) for blocking (see Figure 2, positive and negative peaks and their corresponding blocks have different thresholds). In line to this, in Conclusions, lines 301-303, the phrase ‘when the blocking threshold is closer to the peak of the sinusoidal LFAC waveform, the slower rate of
change of the amplitude allows for a more substantial conduction delay’
makes sense for different fiber diameter however for the same fiber size, the conduction delay remains equal no matter if the thresholds cathodic and anodic are closer or further to the peak amplitude of the blocking waveform. The paragraph requires better explanation to avoid confusion for the reader.

Author Response

Comment 1: Line 41: the sentence mentions three key gaps, however later on there are two enumerations.

Response 1: This was a remnant from a prior draft. The word ‘three’ has been removed to match the two enumerated aims that follow.

Comment 2: 2.1. In-vivo Earthworm Experiment: indicate the length of the worm as performed with the canine vagus nerve and distances of the stimulation, blocking and recording sites.

Response 2: The length of the earthworm and the distances between the stimulation, recording, and blocking sites have been added to the manuscript in section 2.1 for consistency with section 2.2.

Comment 3: 2.1. In-vivo Earthworm Experiment: why did you use a pentapolar cuff for blocking if you only required a cathode and anode?

Response 3: The pentapolar cuff was used as part of early experimental exploration of LFAC waveforms. Alternative configurations, including tripolar and dual-bipolar, were also tested during this stage, though not shown in this manuscript.

Comment 4: For sections 2.1 and 2.2, specify the distances in millimeters for stimulation, blocking, and the upstream and downstream recordings. Providing these details will be valuable for other research groups working with similar in-vivo and in-silico models.

Response 4: Section 2.1 was addressed in Comment 2. For section 2.2, the approximate distances between cuff electrodes (10–20 mm) have now been included in the revised manuscript.

Comment 5: Lines 206-207: It is mentioned that, in the upstream recording cuff, the effect observed before the CAP passes the blocking cuff electrode is likely due to currents escaping the structure of the cuff electrode. To support this statement, have you measured the contribution of the current escaping from the cuff and affecting the membrane potential in the surroundings in your in-silico model? If this is the case, please add corresponding information in the Results section.

Response 5: A measure of the current escaping the current was not done for this analysis. However, this measure has been done by this lab for cuff electrode design by looking at the current passing between the contacts and the current escaping the cuff. This escaping current plays a role in establishing virtual electrodes, a common effect of using cuff electrodes.

Comment 6: Lines 224–226: To improve visualization, consider adding a plot similar to Figures 1 and 2, where the maps are shown alongside the blocking waveform. A zoomed-in view of the waveform, focusing on this small current range, could be beneficial for the reader. Additionally, the phrase ‘As amplitude increased (0.70 mA and 1.0 mA), full block occurred with shorter durations at higher amplitudes,’ ensure clarity in describing the relationship between amplitude and blocking duration, because from Figure 8 G-I, as the current increases in amplitude the full block duration intervals increase.

Response 6:While adding the LFAC waveform to the figure could help illustrate which phase induces block, this was not the primary focus of the figure. The intent of Figure 8 was to show the subthreshold behavior of the action potential (AP), including how changes in waveform frequency or fiber size did not affect the shape of conduction delay prior to block (i.e., the AP “smear” along the x-axis).

Green overlay text has been added to each panel to highlight key distinctions, such as frequency, fiber diameter, conduction velocity, and amplitude. The figures are of sufficient resolution for digital zooming to help interpret these attributes clearly.

Additionally, the original sentence has been corrected: “shorter” has been replaced with “longer” to more accurately reflect that increasing LFAC amplitude prolongs the block window (y-axis duration).

Comment 7: In section 4.3, line 255, is it conduction delay? The reason is that in Figure 8 G-I, as the amplitude of the LFAC increases, the conduction delay does not show a change. Contrary is the time duration of blocking which increases with LFAC amplitude.

Response 7: The section has been rewritten to clarify that LFAC amplitude affects the window in which conduction delay occurs, rather than the total conduction delay itself. The dominant factor in conduction delay magnitude is fiber type, with unmyelinated HYS fibers exhibiting longer delays (~5 ms) than faster-conducting MRG fibers (~1 ms). This difference is shown in Figures 6 and 8 and also reflects trends seen in the earthworm data (Figure 1), where the LGF (slower) AP showed longer delays than the MGF (faster).

Comment 8: In line 289, Figure 6G is not present.

Response 8: This has been corrected to read “Figure 8G” as originally intended.

Comment 9: In section 4.6, lines 291-292, it says that ‘smaller diameter fibers are more sensitive to the local extracellular field…’. This statement is not fully clear considering that small fibers have a larger threshold which can be understood as less sensitive to extracellular fields. Please add more explanation. In addition, the unevenness is also not fully clear in images. See previous comment.

Response 9: The section has been rewritten for clarity. The intention was to convey that smaller, slower fibers exhibit more conduction delay—not that they have lower block thresholds. Larger fibers are indeed more sensitive to extracellular fields in terms of block threshold, but slower fibers exhibit greater subthreshold conduction delay, which was observed both in vivo and in silico.

The unevenness noted in conduction delay was also clarified: this is likely due to geometric differences in myelinated fibers, such as the positioning of nodes of Ranvier relative to the field. The correct pointer to Figure 8G has also been updated to better illustrate this point.

Comment 10: In the manuscript, I couldn’t find any mention on the different thresholds (cathodic and anodic) for blocking (see Figure 2, positive and negative peaks and their corresponding blocks have different thresholds). In line to this, in Conclusions, lines 301-303, the phrase ‘when the blocking threshold is closer to the peak of the sinusoidal LFAC waveform, the slower rate of change of the amplitude allows for a more substantial conduction delay’ makes sense for different fiber diameter however for the same fiber size, the conduction delay remains equal no matter if the thresholds cathodic and anodic are closer or further to the peak amplitude of the blocking waveform. The paragraph requires better explanation to avoid confusion for the reader.

Response 10: This manuscript is intended to be observational and not quantitative due to the nature of the data collected. A more comprehensive analysis of LFAC block thresholds will be forthcoming in a future paper where a better protocol was followed to allow for blocking thresholds to be determined more systematically. 

I want to thank the reviewer for this point as it points out a miscommunication of the authors idea. As was shown in Figure 4 with the canine ex-vivo data, frequency is not a key player in the amount of conduction delay before block. This is further confirmed for unmyelinated HYS fibers in Figure 6 A-C and myelinated MRG fibers in Figure 8 A-C. What was meant by the phrase ‘when the blocking threshold is closer to the peak of the sinusoidal LFAC waveform, the slower rate of change of the amplitude allows for a more substantial conduction delay’ was not the total conduction delay (x-axis) but rather that the conduction delay could be maintained longer (y-axis), without block. This is shown in Figure 6D when compared to 6E and 6F where conduction delay was almost maximized (x-axis) without blocking. At a slightly lower LFAC amplitude conduction delay could be modulated from 0ms delay up to nearly 10ms without a block. This conduction delay without block is partially seen in Figure 8F for the myelinated MRG fibers as the 'uneven' effect where a portion of the LFAC waveform achieves block and the other does not. 

The first paragraph of the conclusion section has been modified to remove this confusion. 

Reviewer 3 Report

Comments and Suggestions for Authors

In this manuscript, the authors demonstrated the impacts of low-frequency alternating current block (LFACb) on action potential conduction through multiple models. These models included: (a) in - vivo experiments conducted on earthworms, which provided real - time physiological responses in a living organism; (b) ex-vivo experiments carried out on canine vagus nerves, enabling a more direct observation of nerve function outside the body while maintaining some physiological characteristics; (c) in-silico simulations employing the Horn, Yoshida, and Schild (HYS) model for unmyelinated fibers, allowing for a theoretical exploration of the electrical behavior in such fibers; and (d) in-silico simulations using the McIntyre, Richardson, and Grill (MRG) model, which further enhanced the understanding of the phenomenon from a different computational perspective. Overall, the study is well designed. The results are clearly presented and strongly support the conclusion.

However, there are still a few minor concerns:

(1) It is recommended that the authors create a schematic figure to comprehensively summarize their research findings.

(2) The authors are advised to quantify their data and perform statistical analyses. 

Author Response

Comment 1: It is recommended that the authors create a schematic figure to comprehensively summarize their research findings.

Response 1: We appreciate the reviewer’s suggestion to create a schematic summarizing the research findings. The data presented in this study demonstrate clear trends in conduction delay and blocking behavior across different fiber types, diameters, and frequencies. However, generating a more comprehensive summary plot that quantitatively captures these trends would require an extensive in-silico study designed specifically to analyze the interactions between these parameters in greater depth.

As this manuscript is focused on general observations of LFACb-induced conduction delay and block, we believe that the current presentation of results—through experimental and computational figures—best reflects the scope of this work. Future studies with a dedicated computational focus could systematically quantify these relationships and provide an opportunity for a more detailed schematic representation of LFACb effects.

Comment 2: The authors are advised to quantify their data and perform statistical analyses.

Response 2: We appreciate the reviewer’s suggestion to quantify the data and perform statistical analyses. This study was designed as an observational investigation into the effect of conduction delay before block, focusing on qualitative trends rather than statistical comparisons. A more comprehensive statistical analysis would require a larger sample size, particularly in animal experiments, to achieve statistical significance. Future studies may incorporate expanded datasets to allow for quantitative validation of these findings.

Round 2

Reviewer 1 Report

Comments and Suggestions for Authors

The authors have addressed my main concerns with their additions to the discussion, however they have not generated the quantifiable data that needs to be in the results.  In addition, although the discussion has been updated, the introduction still does not adequately introduce the reader to the important motivation for the work.  

  • The additions to the discussion are good, but the addition of a reference to DC block in the introduction does not adequately describe the relationship between your waveform and DC block. It still makes it sound like they are two entirely different things.  Also, the paper you cite describes DC block as both cathodic and anodic (via a virtual cathode), which is a property that you leverage with your waveform.  The introduction also refers to LFACb as “safer” without providing any context or reason for this. 

A more appropriate description would be:

Line 37: “Direct current (DC) block has been explored as another alternative, where a sustained DC field induces nerve conduction block without onset activity through sodium channel inactivation [6].”

Line 42: “In contrast, LFACb avoids onset activation entirely by leveraging the properties of DC block.  The inherent charge balancing of the LFACb waveform offers a safer alternative to DC provided high capacitive materials are used to mitigate the larger cyclic charge injection resulting from the use of lower frequencies. This waveform could result in a more versatile approach for applications requiring selective nerve modulation.”

This is really the crux of why you are using this waveform.  It mitigates the onset, but since it is alternating, it is charge balanced.  But the frequency is still low enough that you will exceed the water window with conventional materials.

  • In your time triggered plots, you are already extracting the height of the CAP (plotted as the color, correct?) which you use to determine the amount of block over time. You just need to measure the time it takes to go from no block to complete block.  You mention that “there were no significant differences in the magnitude of the conduction delay”, but you don’t provide any numbers to back this up.  You also report the conduction velocity, so it seems like this is just a matter of creating new plots with the actual numbers. One figure (figure 5) showing an example of the output as a time triggered plot is a good illustration to put the experiment in context, but the rest of the data would be clearer if it were quantified.  Figure 8 is particularly difficult to read.  The objective of the paper is the delay (or conduction velocity depending on how you want to represent it), so it is important to report it in a robust way.  Ideally you would report statistics to show that diameter is significant but frequency is not. 

Author Response

Comment 1: The authors have addressed my main concerns with their additions to the discussion, however they have not generated the quantifiable data that needs to be in the results.  In addition, although the discussion has been updated, the introduction still does not adequately introduce the reader to the important motivation for the work. 

Response 1: The current study was designed as a qualitative investigation into the effects of LFACb on conduction delay and block onset across multiple models. While the results do not include quantified data or statistical comparisons due to limitations in resolution and sample size in the experimental datasets, this paper establishes key observable trends—particularly the subthreshold delay phenomenon—that justify and inform future studies focused on systematic quantification. A follow-up study is currently underway involving in-vivo rat experiments with more controlled waveform delivery, fiber-type-specific targeting, and geometry comparisons to allow for robust statistical analysis.

The introduction has been revised to more clearly communicate the clinical relevance of conduction slowing as a mechanism of interest, especially in contexts where signal delay, rather than full block, may provide therapeutic value. Examples now include its potential application in reflex timing alignment, tremor desynchronization, and graded neuromodulation strategies in FES. The revised text also reinforces the rationale for studying LFACb as a waveform that offers not just safer alternatives to DC and kHFAC, but a broader spectrum of modulatory control.

Comment 2: The additions to the discussion are good, but the addition of a reference to DC block in the introduction does not adequately describe the relationship between your waveform and DC block. It still makes it sound like they are two entirely different things.  Also, the paper you cite describes DC block as both cathodic and anodic (via a virtual cathode), which is a property that you leverage with your waveform.  The introduction also refers to LFACb as “safer” without providing any context or reason for this. 
A more appropriate description would be:

Line 37: “Direct current (DC) block has been explored as another alternative, where a sustained DC field induces nerve conduction block without onset activity through sodium channel inactivation [6].”

Line 42: “In contrast, LFACb avoids onset activation entirely by leveraging the properties of DC block.  The inherent charge balancing of the LFACb waveform offers a safer alternative to DC provided high capacitive materials are used to mitigate the larger cyclic charge injection resulting from the use of lower frequencies. This waveform could result in a more versatile approach for applications requiring selective nerve modulation.”

This is really the crux of why you are using this waveform.  It mitigates the onset, but since it is alternating, it is charge balanced.  But the frequency is still low enough that you will exceed the water window with conventional materials.

Response 2: The introduction has been revised to clarify the mechanistic relationship between LFACb and DC block. Both approaches rely on sodium channel inactivation to achieve conduction block. However, LFACb delivers this effect through a time-varying, sinusoidal waveform that is charge-balanced and initiated at zero crossings. This design fundamentally avoids the unintended excitations that are characteristic of both DC and kHFAC block.

In particular, while DC block can induce make-and-break activation at the onset and offset of the waveform—as demonstrated in simulations by Bhadra and Kilgore (2004)—LFACb, when properly timed, does not suffer from this problem. Similarly, kHFAC block is known to cause onset activation due to its abrupt application of high-frequency current. LFACb, by contrast, introduces current gradually and symmetrically, avoiding these transient excitatory responses altogether. In this sense, LFACb functions as a "pure" block waveform, offering a unique ability to block conduction without triggering any form of activation during application or cessation.

The reviewer also noted that DC block can involve anodic effects via a virtual cathode, and LFACb, by nature of its alternating waveform, shares this capability. The biphasic nature of LFACb supports both cathodic and anodic phases distributed in time and space, enabling flexible orientation-dependent modulation of nerve fibers.

Regarding safety, LFACb is considered safer than constant DC because its alternating, charge-balanced waveform avoids net charge buildup that could lead to electrode degradation or tissue damage. Although its low frequency increases the risk of exceeding the water window, this can be mitigated by using high-capacitance electrode materials (e.g., platinum black or PEDOT), which expand the usable charge injection range. These safety considerations and mechanistic distinctions have been incorporated into the revised introduction to better convey the motivation for using LFACb.

Comment 3: In your time triggered plots, you are already extracting the height of the CAP (plotted as the color, correct?) which you use to determine the amount of block over time. You just need to measure the time it takes to go from no block to complete block.  You mention that “there were no significant differences in the magnitude of the conduction delay”, but you don’t provide any numbers to back this up.  You also report the conduction velocity, so it seems like this is just a matter of creating new plots with the actual numbers. One figure (figure 5) showing an example of the output as a time triggered plot is a good illustration to put the experiment in context, but the rest of the data would be clearer if it were quantified.  Figure 8 is particularly difficult to read.  The objective of the paper is the delay (or conduction velocity depending on how you want to represent it), so it is important to report it in a robust way.  Ideally you would report statistics to show that diameter is significant but frequency is not. 

Response 3:The magnitude of block was not quantified in either the in-vivo earthworm or ex-vivo canine vagus nerve experiments. This manuscript is focused on the qualitative behavior of LFACb, specifically the observation of conduction delay prior to full block onset. No measurements were taken to quantify signal attenuation over time, and the term "significant" has been removed throughout the manuscript to avoid implying statistical testing.

The experiments were exploratory in nature and limited in biological replicates, making statistical comparisons impractical. Additionally, the low stimulation rates used in the in-vivo (4.4 Hz) and ex-vivo (12 Hz) models limited the temporal resolution of LFAC phase sampling, further constraining the ability to extract consistent delay metrics across different frequencies or time windows.

Importantly, the exact distance between stimulation and recording electrodes in the biological experiments was not recorded—a limitation of the experimental design that we acknowledge. Without this information, it is not possible to convert conduction delay to velocity with confidence. This constraint led us to display triggered time delay uniformly across all figures (rather than conduction velocity) to maintain consistency between biological and in-silico models. While conduction velocity could be reported for the in-silico simulations, using mixed metrics across figures could introduce confusion, especially since the biological data would rely on estimates.

A follow-up study is currently underway that addresses these limitations, using a more controlled in-vivo model to statistically evaluate block thresholds across multiple electrode geometries. That study is specifically designed for quantification and hypothesis testing. The current work serves as a foundational study to introduce and characterize the subthreshold conduction delay effect of LFACb across biological and computational models.

We acknowledge the reviewer’s concern regarding the readability of Figure 8. To address this, we have revised the figure to include green annotations directly on each subplot, highlighting the key experimental parameters such as LFACb amplitude (in panels G–I), frequency (A–C), and conduction velocity (D–F). These visual cues are intended to guide the reader’s attention to the most relevant comparisons across conditions.

While the figure is scaled for fitting within the page layout, it is rendered at high resolution to support zooming and inspection in digital formats, where finer details can be easily examined. We believe these adjustments improve interpretability while preserving consistency with the manuscript’s visual structure.

Reviewer 2 Report

Comments and Suggestions for Authors

Dear authors, thank you for your replies and for adding the proposed suggestions to the manuscript. I don't have further comments.

Round 3

Reviewer 1 Report

Comments and Suggestions for Authors

The authors have addressed my comments